# Mild Head Trauma (MHT) and Antiplatelet Therapy. Reply to Lorenzati et al. Comment on “Savioli et al. Mild Head Trauma: Is Antiplatelet Therapy a Risk Factor for Hemorrhagic Complications? *Medicina* 2021, *57*, 357”

**DOI:** 10.3390/medicina57090889

**Published:** 2021-08-27

**Authors:** Gabriele Savioli, Iride Francesca Ceresa, Sabino Luzzi, Alice Giotta Lucifero, Ginevra Cambiè, Federica Manzoni, Lorenzo Preda, Giovanni Ricevuti, Maria Antonietta Bressan

**Affiliations:** 1Emergency Department, Fondazione IRCCS Policlinico San Matteo, 27100 Pavia, Italy; irideceresa@gmail.com (I.F.C.); mita.bressan@gmail.com (M.A.B.); 2PhD School in Experimental Medicine, Department of Clinical-Surgical, Diagnostic and Pediatric Sciences, University of Pavia, 27100 Pavia, Italy; 3Neurosurgery Unit, Department of Clinical-Surgical, Diagnostic and Pediatric Sciences, University of Pavia, 27100 Pavia, Italy; sabino.luzzi@unipv.it (S.L.); alicelucifero@gmail.com (A.G.L.); 4Neurosurgery Unit, Department of Surgical Sciences, Fondazione IRCCS Policlinico San Matteo, 27100 Pavia, Italy; 5Medicina Generale, Fondazione IRCCS Policlinico San Matteo, 27100 Pavia, Italy; ginevracambie@gmail.com; 6Health Promotion—Environmental Epidemiology Unit, Hygiene and Health Prevention Department, Health Protection Agency, 27100 Pavia, Italy; federica.manzoni@unipv.it; 7Radiology Unit, Fondazione IRCCS Policlinico San Matteo, 27100 Pavia, Italy; l.preda@smatteo.pv.it; 8Department of Drug Science, University of Pavia, 27100 Pavia, Italy; giovanni.ricevuti@unipv.it; 9Saint Camillus International University of Health Sciences, 00152 Rome, Italy

We read your data with interest, and we truly appreciate the similar experience.

Even though minor head injury (MHI) is one of the most common emergency room scenarios in Italy and abroad, it has poor guidelines. Minor head injury is defined as a patient with a history of loss of consciousness, amnesia, or disorientation, and a Glasgow Coma Scale (GCS) score between 13 and 15 [1,2].

Anticoagulant therapy and patients’ outcomes

Anticoagulant therapies are suitable for multiple indications, and they were applied in several cohort studies [3,4,5,6,7,8,9,10,11,12,13,14,15]. The use of these drugs aggravates the risk of traumatic intracranial injury and influences the clinical outcomes after MHI and blunt head trauma [16,17,18,19,20,21]. However, there are no clear guidelines about the management of patients treated with vitamin K antagonist (VKA), new oral direct anticoagulants (DOACs), and antiplatelet agents (APT), unless they underwent a head CT scan upon admission to the ED, and after 24–48 h. As the use of DOACs is becoming more frequent, there has been a more tangible interest in this topic, as well as prospective studies and literature reviews [22,23,24,25,26,27,28,29,30,31]. The research has underlined that the treatment with DOACs has a higher safety profile after an MHI, rather than VKAs, such as in elderly patients [32,33]. Evidence shows that DOACs have a good prophylactic effect against thromboembolism and reduce some side effects, such as intracerebral hemorrhage [34,35,36,37].

Furthermore, DOACs result as easier to work with thanks to the unnecessary monitoring of the anticoagulation activity. On the other hand, they are not so commonly used due to the lack of a known antidote [38].

Recent research, however, has shown a higher risk for ICH, ICH progression, or death, as well as and regarding hematoma expansion, mortality, and neurosurgical treatment in patients being treated with DOACs compared to VKAs [27,28,39,40].

We conducted a study to assess the bleeding risk profiles of patients admitted to the ED for MHI under DOACs and traditional VKAs [16]. The primary endpoint was to dictate the difference between the recurrency and complications of post-traumatic ICH following MHI. The secondary objective was to assess the need for surgery. Intrahospital mortality rates, ED revisit rates, and the volume of ICH were also analyzed. The result of the study was that patients on DOACs treatment had a similar outcome to patients that were not taking any medications. On the other hand, patients following VKAs therapy showed twice the prevalence of ICH compared to the control group (patients not undergoing any treatment) or patients on DOACs, without any differences regarding the need for neurosurgical intervention. It is important to mention that we also obtained similar results when it was adjusted for different age groups.

Anti-platelet therapy and patients’ outcomes

When navigating through the literature, it is unclear whether APT predisposes to bleeding in MHT. It is well-known that anticoagulant therapy increases the risk for bleeding but, on the other hand, the use of antiplatelet drugs is still controversial. Most reports suggest that patients using APTs have the tendency to bleed, but some clarification as to whether this represents a risk factor for hemorrhagic complications should be made. The older the patients are, the higher the risk of bleeding. From our experience, the link between APTs treatment and the number of MHI patients with associated ICH was not statistically significant. However, we agree with our colleagues that the patients have a higher need for neurosurgical treatment, longer hospitalization, and more frequent admissions to the ED.

Trauma-induced coagulopathy and patients’ outcomes

MHI followed by a TIC usually occurs within minutes of head trauma [22,41]. It can be understood as a release of substances triggered by brain damage through the affected blood–brain barrier (BBB). Through ischemic and inflammatory lesions, the trauma can increase the permeability of the BBB, the process carried out by the endothelial cell junction proteins, such as claudins [40] and junctional adhesion molecules [10,42]. This increase in permeability causes fluid leakage and consequent cerebral edema. The cerebral edema eventually contributes to releasing some substances responsible for triggering systemic coagulopathy. It is believed that brain-derived cellular micro-vesicles (BDMV) may play a role as a diffusion factor and as a causal factor.

A recent study shows, using mouse models, BDMV’s rapid release into the circulation associated with a state of systemic hypercoagulability, which rapidly evolved into consumption coagulopathy [42]. It may seem that the strong procoagulant effect may be due to an expression of the large tissue factor and phosphatidylserine, and the infusion of these substances leads to a hypercoagulable state in no-trauma mice [42].

Some reports detected fibrinogen degradation product even before the alteration of prothrombin time (PT) and partial thromboplastin time (PTT). Their peaks were compared at about 3–6 h post-TBI, and they hypnotize an early conversion from a hypercoagulable to a hypocoagulable state [40]. However, the role of head trauma in regulating the changes in fibrinolysis and platelet function still needs to be understood. Regarding this last objection, a rather low platelets counts is seen to be a characteristic of patients with TBI, but their activation leads to a pro-coagulant condition [42].

ED patient management, observation

There is no difference in the duration of the observation between patients undergoing VKAs or DOACs—it has to last 24 h. This period leads to a longer sojourn in the ED and higher use of resources. Even if it is known that the incidence of late ICH, after a negative control CT scan, is between 0% and 7%, the Italian guidelines state that all anticoagulated patients should have an observational window of 24 h and a second CT before discharge, except for elder patients, where a longer stay in the ED can lead to various risks (delirium, nosocomial infections, etc.), and a discharge after the time of observation or community observation can be performed with all the elucidations to the caregiver about possible red flags.

Furthermore, a study showed higher ICH risk on the second CT scan in patients undergoing warfarin, suggesting that VKAs might be more often link to delayed bleeding than DOACs. However, we agree with our colleagues that further studies have to be performed.

From our experience, the diagnosis of ICH is made mostly (95.45%) at the first CT within the first 6 h after the head trauma. Nevertheless, there is a significant percentage of patients (4.55%) for which the diagnosis is made with the second CT scan (24 h later). Specifically, a positive CT scan 24 h later was detected in 1.28% of patients on DOAC and 3.2% of patients on VKAs, underlining the value of the observational period for these patients. The percentage of positive second CT scans is similar to that in previous studies [24,41,43,44,45,46,47,48,49]. Moreover, these patients did not require surgery.

We believe, in line with other authors, that patients with a negative first CT scan and without neurological deterioration can be discharged after a clear period of in-ward observation and without a second CT scan, especially for patients on DOACs.

Regarding the discontinuation and resumption of anticoagulant therapy with TAO/NAO, it will substantially depend on two sets of factors: whether the patient has bled and why they take anticoagulant therapy. The opinion of the authors is as follows: In case of non-bleeding, continue therapy. In case of intracranial bleeding, it is necessary to evaluate the risk/benefit ratio of the suspension by scheduling a multidisciplinary evaluation at 30 days with a neurosurgical follow-up visit and a specialist visit regarding the reasons for which anticoagulant therapy was undertaken (previous pulmonary embolisms, atrial fibrillation, etc.). In this regard, we recall that the resumption of therapy should also depend on the type of approach used in the evaluation of the head injury, and in particular, on the possible execution of the second CT scan. If anticoagulant therapy is administered for atrial fibrillation, and in the event that the second scan is postponed, with the first scan being negative and the patient having no symptoms, it is the authors’ opinion that the anti-coagulant therapy can be resumed after a break of one week. This is in consideration of the fact that a small percentage of patients may have bleeding, albeit the percentage is insignificant. If the patient is valvular or has recent pulmonary thrombus embolism, anticoagulant therapy with vitamin K antagonists must be resumed within 48 h of the trauma in order not to increase the thrombotic risk too much.

The overall consideration is that more data are needed, particularly from multicentric analyses, to suggest new ED management guidelines.

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
