# Peer review of "Mild Head Trauma (MHT) and Antiplatelet Therapy. Reply to Lorenzati et al. Comment on “Savioli et al. Mild Head Trauma: Is Antiplatelet Therapy a Risk Factor for Hemorrhagic Complications? Medicina 2021, 57, 357”"

_medicina, 2021, doi:10.3390/medicina57090889_

Round 1

Reviewer 1 Report

Well written response. I especially appreciate the update in knowledge provided here beyond the original paper, and also regarding the nuances of management as of course some patients will have increase in ICH when on VKA/DOACs after TBI. The other important note, as stated, is that the majority of patients do not need surgical intervention.

The remaining question (or future direction), if able to be answered, is when these DOACs/VKAs should be resumed. E.g. should these be held and not resumed until a follow-up appointment in Neurosurgery clinic? Worth stating and/or discussing.

Minor comments:

- Line 119: "In line with the other authors..." rather than "lining up"

Author Response

response to reviewer point by point

Well written response. I especially appreciate the update in knowledge provided here beyond the original paper, and also regarding the nuances of management as of course some patients will have increase in ICH when on VKA/DOACs after TBI. The other important note, as stated, is that the majority of patients do not need surgical intervention.

The remaining question (or future direction), if able to be answered, is when these DOACs/VKAs should be resumed. E.g. should these be held and not resumed until a follow-up appointment in Neurosurgery clinic? Worth stating and/or discussing.

Dear Reviewer, we thank you for your contribution and your suggestions that contribute to improving our response

We therefore propose to add the following sentence to the manuscript:

" Regarding the discontinuation and resumption of anticoagulant therapy with TAO/NAO will depend substantially on two sets of factors: whether the patient has bled and from why he takes anticoagulant therapy. The opinion of the authors is as follows: in case of non-bleeding continue therapy. In case of intracranial bleeding, it is necessary to evaluate the risk/benefit ratio of the suspension by scheduling a multidisciplinary evaluation at 30 days with a neurosurgical follow-up visit and a specialist visit regarding the reasons for which anticoagulant therapy was undertaken (previous pulmonary embolisms, atrial fibrillation ...). In this regard, we remind that the resumption of therapy should also depend on the type of approach used in the evaluation of the head injury, and in particular on the possible execution of the second CT scan. Ifantcoagulant therapy is given for atrial fibrillation, and in the event that the second scan is postponed, the first being negative and the patient having no symptoms, it is the authors' opinion that the anti-coagulant therapy can be resumed after a stop of a week. this considering the fact that a small percentage of patients may have bled, albeit insignificantly. If the patient is valvular or has recent pulmonary thrombus embolism, anticoagulant therapy with vitamin K antagonists must be resumed within 48 hours of the trauma in order not to increase the thrombotic risk too much."

Minor comments:

- Line 119: "In line with the other authors..." rather than "lining up"

dear reviewer, we thank you for the contribution we welcome